# UNIVERSAL IMAGE RESTORATION PRE-TRAINING VIA DEGRADATION CLASSIFICATION

**JiaKui Hu**[1,2,3], **Lujia Jin**[4], **Zhengjian Yao**[1,2,3], **Yanye Lu**[1,2,3*]
[1]Institute of Medical Technology, Peking University Health Science Center, Peking University
[2]Biomedical Engineering Department, College of Future Technology, Peking University
[3]National Biomedical Imaging Center, Peking University
[4]China Mobile Research Institute

## ABSTRACT

This paper proposes the Degradation Classification Pre-Training (DCPT), which enables models to learn how to classify the degradation type of input images for universal image restoration pre-training. Unlike the existing self-supervised pre-training methods, DCPT utilizes the degradation type of the input image as an extremely weak supervision, which can be effortlessly obtained, even intrinsic in all image restoration datasets. DCPT comprises two primary stages. Initially, image features are extracted from the encoder. Subsequently, a lightweight decoder, such as ResNet18, is leveraged to classify the degradation type of the input image solely based on the features extracted in the first stage, without utilizing the input image. The encoder is pre-trained with a straightforward yet potent DCPT, which is used to address universal image restoration and achieve outstanding performance. Following DCPT, both convolutional neural networks (CNNs) and transformers demonstrate performance improvements, with gains of up to 2.55 dB in the 10D all-in-one restoration task and 6.53 dB in the mixed degradation scenarios. Moreover, previous self-supervised pretraining methods, such as masked image modeling, discard the decoder after pre-training, while our DCPT utilizes the pre-trained parameters more effectively. This superiority arises from the degradation classifier acquired during DCPT, which facilitates transfer learning between models of identical architecture trained on diverse degradation types. Source code and models are available at `https://github.com/MILab-PKU/dcpt`.

## 1 INTRODUCTION

Image restoration is the task of using models to improve low-quality (LQ) images into high-quality (HQ) images. Recently, deep learning based methods (Li et al., 2022; Potlapalli et al., 2023; Ai et al., 2024; Luo et al., 2023b; Zheng et al., 2024; Guo et al., 2025) have shown better performance and efficiency than traditional methods (Buades et al., 2005; Dabov et al., 2007; Yang et al., 2010) in scenarios with variable and mixed degradation. Prevailing methods use degradation representations of the input image as discriminative prompts for universal restoration, such as gradients (Ma et al., 2020), frequency (Ji et al., 2021), additional parameters (Potlapalli et al., 2023) and abstracted features compressed by neural networks (Li et al., 2022; Ai et al., 2024; Luo et al., 2023b; Zheng et al., 2024; Wang et al., 2024). Subsequently, these degradation representations serve as prompts for advanced generative models that have been either fine-tuned or trained for the universal image restoration task. Although such methods attain high performance through the use of precise and effective prompts, they fail to utilize the latent prior information embedded within the restoration model itself.

Self-supervised pre-training strategies (Devlin et al., 2019; Radford et al., 2019; Brown et al., 2020; Grill et al., 2020; Caron et al., 2021; Chen et al., 2021c; He et al., 2022; Xie et al., 2022) activate latent discriminant information within neural networks, thereby facilitating the acquisition of universal input signal representations and making the pre-trained model suitable for downstream tasks. Contrastive learning (Chen et al., 2020b; He et al., 2020) discover representations by maximizing agreement across multiple augmented views of the same sample using contrastive loss (Oord et al., 2018), obtaining features with fine-grained discriminant information (Chen et al., 2021c). Masked image modelling (MIM) (He et al., 2022; Xie et al., 2022; Tian et al., 2022) extends BERT's (Devlin et al., 2019) success from language to vision transformers and CNNs. MIM presents a challenging image reconstruction task through a staggeringly high mask ratio (60 ~75%), forcing the model to excavate

---

*Corresponding author, yanye.lu@pku.edu.cn

the intrinsic distribution of images. With GPT's (Radford et al., 2019; Brown et al., 2020) success in language generation, related methods (Chen et al., 2020a) are used in image generation. However, self-supervised pre-training for image restoration is scarce (Liu et al., 2023a; Chen et al., 2023a), limited to single-task applications, and does not leverage universal representations from large-scale pre-training. It is imperative to find the discriminative information within the restoration models and employ pre-training strategies to augment it. This will create a pre-trained restoration model capable of addressing universal restoration tasks, including multi-task (all-in-one), single-task, and mixed degradation scenarios.

In this paper, we propose that the degradation classification capability is an often overlooked yet powerful discriminative information inherent in restoration models, and we employ it in the pre-training for universal image restoration tasks. We first explore the degradation classification capabilities intrinsic to established classic (Liang et al., 2021; Chen et al., 2022; Zamir et al., 2022) and all-in-one (Potlapalli et al., 2023) image restoration architectures. It is striking to discover that the randomly initialized models exhibit a preliminary ability to classify degradation, which is further enhanced after all-in-one restoration training, enabling them to discern previously unseen degradation types. This observation indicates that the network inherently possesses degradation identification prowess, which is progressively refined through the training stage.

Inspired by this observation, we propose the Degradation Classification Pre-Training (**DCPT**) framework for universal image restoration tasks. This approach endows the models with robust prior information on degradation comprehension via conducting the degradation classification learning phase beforehand, thereby augmenting the model's universal restoration abilities. Specifically, DCPT follows an encoder-decoder design, where the encoder converts input images into abstract features, and a lightweight decoder classifies the degradation type based on the encoder's output features. The pre-trained encoder initializes a restoration model during fine-tuning for downstream tasks, leading to significant performance gains. Experimental results demonstrate that the DCPT framework significantly optimizes the performance of various architectures in the realm of restoration tasks, encompassing ***all-in-one*** and ***mixed degradation*** conditions. Moreover, the pre-trained decoder facilitates transfer learning between models of identical architecture trained on different degradation, thereby enhancing the model's ability to generalize across different types of degradation.

## 2 RELATED WORK

**Image Restoration.** Recent deep-learning methods (Liang et al., 2021; Geng et al., 2021; Chen et al., 2021b; Mei et al., 2021; Zamir et al., 2022; Tu et al., 2022; Chen et al., 2022; 2023b; Yawei Li et al., 2023; Jin et al., 2024) have consistently shown better performance and efficiency compared to traditional methods in single-task image restoration. The proposed neural networks are based on convolutional neural networks (CNNs) (LeCun et al., 2015) and Transformers (Vaswani et al., 2017). CNNs (Lim et al., 2017; Zhang et al., 2018; Chen et al., 2021b; Mei et al., 2021; Chen et al., 2022; Niu et al., 2020) are highly effective at processing local information in images, while Transformers (Liang et al., 2021; Zamir et al., 2022; Wang et al., 2022; Yawei Li et al., 2023; Chen et al., 2023b) excel at leveraging the local self-similarity of images by utilizing long-range dependencies. However, these methods train respective models for each task, even datasets under the same degradation (Chen et al., 2022; Yawei Li et al., 2023; Cui et al., 2022). This renders a significant proportion of methods incapable of effectively addressing the diversities inherent in image restoration (Li et al., 2020).

**Universal Image Restoration** is born for this. It requires a single model to address numerous degradation. In the early universal restoration methods, different tasks are handled by decoupling learning (Fan et al., 2019) or adopting different encoder (Li et al., 2020) or decoder head (Chen et al., 2021a). These methods necessitate that the model explicitly assess degradation and select distinct network branches to cope with different degradation. Recently, AirNet (Li et al., 2022) uses MoCo (He et al., 2020) while IDR (Zhang et al., 2023) creates various physical degradation models to learn degradation representations for all-in-one image restoration. PromptIR (Potlapalli et al., 2023) involves additional parameters via dynamic convolutions to achieve universal image restoration without embedded features. DACLIP (Luo et al., 2023b), MPerceiver (Ai et al., 2024), and DiffUIR (Zheng et al., 2024) use external large models (Radford et al., 2021; Van Den Oord et al., 2017; Esser et al., 2021) or generative prior to achieve higher performance and handle more tasks. OneRestore (Guo et al., 2025) focuses on mixed degradation restoration and offers a benchmark for it. It appears that integrated models must necessitate the regulation of external parameters (Potlapalli

et al., 2023), physical models (Zhang et al., 2023), human instructions (Marcos V. Conde, 2024), and the abstract high-dimension feature extracted by large neural networks (Li et al., 2022; Ai et al., 2024; Luo et al., 2023b; Zheng et al., 2024).

**Self-supervised Pre-training** is a technique that allows the network to learn the distribution (Devlin et al., 2019) or intrinsic prior concealed in input samples and use it to improve the performance in downstream tasks. In computer vision, it is mainly divided into two schools: Contrastive Learning (CL) (Chen et al., 2020b; He et al., 2020) and Masked Image Modeling (MIM) (He et al., 2022; Xie et al., 2022). CL aligns features from positive pairs, and uniforms the induced distribution of features on the hypersphere (Wang & Isola, 2020). MIM learns how to create before learning to understand (Xie et al., 2022). However, it is difficult to extend to other architectures (Tian et al., 2022; Gao et al., 2022; Yao et al., 2025) and discards the decoder during downstream tasks, resulting in inconsistent representations between pre-training and fine-tuning (Han et al., 2023). Recently, many pre-training methods (Chen et al., 2021a; 2023b; Li et al., 2023a) for restoration have been proposed. Unfortunately, these methods only use larger dataset to train larger models under single-degradation settings. The only self-supervised pre-training method (Liu et al., 2023a) for image restoration works well in high-cost tasks but is inappropriate for low-cost tasks like image denoising.

## 3 DCPT: LEARN TO CLASSIFY DEGRADATION

### 3.1 MOTIVATION

When training a single network for the all-in-one image restoration task, the model is expected to learn effective solutions for various degradation and to decide how to restore the input image autonomously. We argue that *image restoration models inherently have the capacity to differentiate between various degradation, and this capacity can be further enhanced through restoration training*.

We perform preliminary experiments for verification. The output feature before the restoration head is extracted, and a k-nearest neighbour (kNN) classifier is employed to classify five degradation types, including haze, rain, Gaussian noise, motion blur, and low-light, of the input image based on this feature. We experiment with both classic image restoration networks (Liang et al., 2021; Zamir et al., 2022; Chen et al., 2022) and a dedicated all-in-one task network (Potlapalli et al., 2023), evaluating both randomly initialized models and those trained on the three distinct (3D) degradation (haze, rain, Gaussian noise) all-in-one image restoration task. It is pertinent to mention that the five target categories for classification encompass the three degradation utilized in the training phase.

| Methods | NAFNet | SwinIR | Restormer | PromptIR |
|---|---|---|---|---|
| Acc. on Random initialized (%) | $52 \pm 1$ | $64 \pm 4$ | $71 \pm 4$ | $55 \pm 3$ |
| Acc. on 3D all-in-one trained 200k iterations (%) | $90 \pm 5$ | $92 \pm 6$ | $93 \pm 3$ | $93 \pm 5$ |
| Acc. on 3D all-in-one trained 400k iterations (%) | $94 \pm 4$ | $95 \pm 4$ | $95 \pm 4$ | $95 \pm 4$ |
| Acc. on 3D all-in-one trained 600k iterations (%) | $94 \pm 5$ | $95 \pm 4$ | $97 \pm 2$ | $95 \pm 4$ |

Table 1: Degradation classification accuracy. The results are averaged under five random seeds.

The results are presented in Table 1 and Figure 1. It can be seen that randomly initialized models can achieve 52 ~ 71 % degradation classification accuracy. After the 3D all-in-one training, models achieve an accuracy of 94% or higher in classifying degradation, including unseen ones. Figure 1 plots the T-SNE results of PromptIR's (Potlapalli et al., 2023) features on five degradation after random initialization and 3D all-in-one training which provides a more natural depiction of the restoration model's classification ability in terms of input image's degradation.

These results lead to three interesting conclusions: **(1)** *Randomly initialized models demonstrate an inherent capability to classify degradation.* **(2)** *Models trained on the all-in-one task exhibit the ability to discern unkown degradation.* **(3)** *There is a degradation understanding step in the early training of the restoration model.* This indicates that the image restoration model inherently possesses the capacity to classify various degradation types. Moreover, training models on the all-in-one image restoration task will further enhance this ability. During the training process of the all-in-one restoration model, it is speculated that while the model is tasked with restoration, it is also simultaneously trained to discern the type of degradation present in the input image. These findings

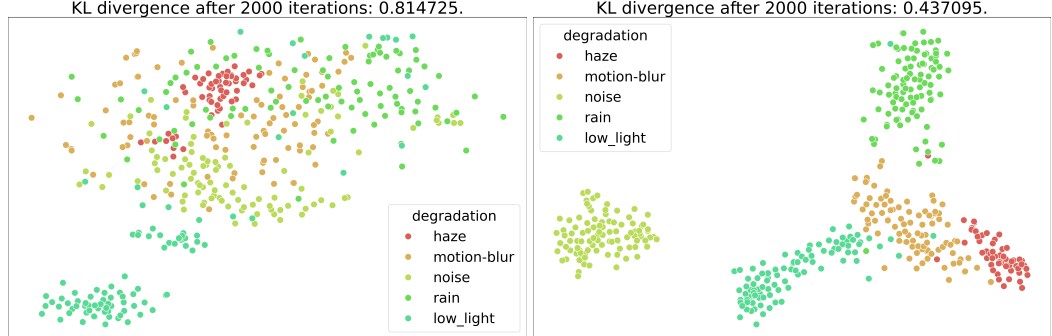

Figure 1: The T-SNE results of randomly initialized PromptIR's feature (left) and all-in-one trained PromptIR's feature (right).

furnish the rationale for our motivation: To ensure superior restoration performance, it is imperative that the restoration model attains sufficient degradation classification capabilities before training.[1]

## 3.2 METHODS

Based on this plain and simple idea, we propose the Degradation Classification Pre-Training (DCPT). DCPT consists of an encoder which comprises restoration models (Liang et al., 2021; Zamir et al., 2022; Chen et al., 2022; Potlapalli et al., 2023) without their restoration modules, and a decoder which classifies the degradation of input images based on the encoder's features. In the degradation classification (DC) stage, given an input image $x_{degrad}$ with degradation $D_{gt}$, encoder's feature $F$ is fed into the decoder. Since the decoder's role is classification rather than image reconstruction, its design draws inspiration from classic classification networks (He et al., 2016), rather than the architecture commonly found in classic autoencoder-style pre-training methods (He et al., 2022). In the generation stage, encoder's feature $F$ is fed into the restoration module to preserve encoder's generation ability. Figure 2 illustrates this overall pipline.

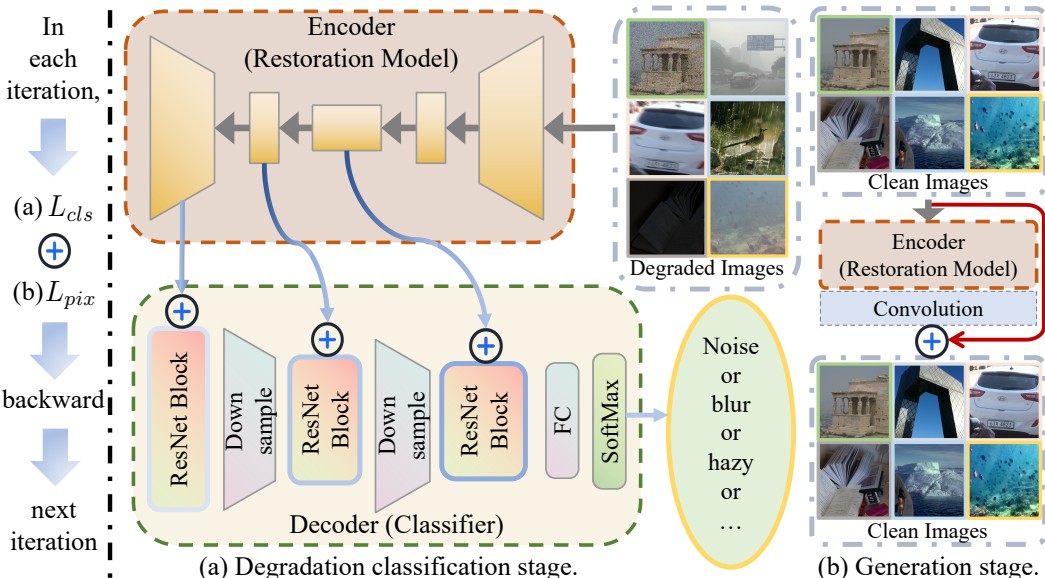

Figure 2: DCPT follows an encoder-decoder design. The encoder refers to a restoration network, and the decoder is a degradation classifier. DCPT consists of two stages. **In each training iteration**, (a) degradation classification stage and (b) generation stage occur **performed alternately**. $L_{cls}$ and $L_{pix}$ are the losses incurred during stage (a) and stage (b). After DCPT, the encoder is fine-tuned for downstream restoration tasks.

**Extract multi-level features.** To achieve more effective degradation classification, it's crucial to extract features from deeper layers that contain richer high-level semantic information (Cai et al., 2023). However, image restoration models typically adhere to the design concept of residual

---

[1]We perform an experimental verification of this motivation in Appendix A.3.

learning (Zhang et al., 2017). The sole reliance on features from the deepest layer for the loss function calculation may result in gradient vanishing in the shallower layers, due to the loss of the encoder's long residual connections (Zhang et al., 2017) during feature extraction. To achieve a balance, features are extracted from each block in the latter part of the encoder. We define these extracted features as $\{F_i\}, i \in 1, \cdots, \frac{l}{2}$, where $l$ is the number of the blocks in the network.

$$\{F_i\} = \text{Encoder}(x_{degrad}). \tag{1}$$

**Degradation Classification.** After multi-level feature extraction, we feed $\{F_i\}$ into a lightweight decoder to classify the degradation of the input images. The details of decoder architecture is shown in Figure 2 (a). To aggregate the extracted features better, it is necessary to perform scaling (Luo et al., 2023a; Yao et al., 2024; Hu et al., 2024) to features $\{F_i\}$. The scaling coefficient $\{\omega_i\}$ is learnable. Then, scaled feature $F'_i = \omega_i F_i$ is plugged into the $i$-th last block in ResNet18 to classify degradation. For stabling the training process (Liu et al., 2022), we replace the normalization layers in decoder from BatchNorm (Ioffe & Szegedy, 2015) to LayerNorm (Ba et al., 2016).

$$D_{pred} = \text{Decoder}(F'_1, F'_2, \cdots, F'_{\frac{l}{2}}). \tag{2}$$

It is crucial to note that the challenge in obtaining image restoration data (Li et al., 2023b) results in an imbalance in the number of datasets representing different types of degradation. For example, the deraining dataset Rain200L (Yang et al., 2017) comprises only 200 images, whereas the dehazing dataset RESIDE (Li et al., 2018) encompasses 72,135 images. This imbalance poses a significant long-tail challenge in classifying degradation. To address this issue, we employ Focal Loss (Lin et al., 2017) as the loss function for long-tail degradation classification.

$$L_{cls} = \text{Focal Loss}(D_{pred}, D_{gt}). \tag{3}$$

**Preserve generation ability.** It is vital to recall that the objective of the encoder is to restore low-quality images into higher-quality ones. Ensuring that the pre-trained encoder maintains its ability to generate is important. Similar to the training process of classic generative methods (Kingma & Welling, 2013), a convolution is added after the encoder to enable it to reconstruct $\hat{x}_{clean}$ from the feature $F_l$, as depicted in Figure 2 (b). The overall loss function of DCPT is as follows:

$$L_{total} = \alpha L_{pix} + L_{cls} = \alpha ||x_{clean} - \hat{x}_{clean}||_1 + \text{Focal Loss}(D_{gt}, D_{pred}), \tag{4}$$

where $\alpha$ is 1 by default, and $\hat{x}_{clean} = \text{Convolution}(F_l)$.

**Divide one pre-training iteration into two alternating stages.** During the DCPT, both loss functions $L_{pix}$ and $L_{cls}$ necessitate the utilization of features generated by the encoder. Concurrently executing the degradation classification stage and the generation stage would result in the encoder receiving two distinct gradient flows simultaneously, a scenario that is impractical. To address this, we alternate between these two stages within one pre-training iteration.

### 3.3 DC-GUIDED TRAINING

Unlike previous pre-training methods (He et al., 2022; Liu et al., 2023a), the decoder is not discarded after pre-training. This leads to DC-guided training for cross-degradation transfer learning.

As illustrated in Figure 3, it is necessary to ensure that the feature of the restoration model can identify the degradation of the input image while restoring clean image $\hat{x}_{restore}$. In DC-guided training, decoder's frozen parameters are set up by DCPT.

The overall loss function in DC-guided training is:

$$\begin{aligned} L_{guide} &= \alpha L_{pix} + L_{cls} \\ &= \alpha ||x_{gt} - \hat{x}_{restore}||_1 \\ &\quad + \text{Cross Entropy Loss}(D_{gt}, D_{pred}), \end{aligned} \tag{5}$$

where $\alpha$ is 1 by default.

Cross Entropy Loss is employed as the classification loss for DC-guided training due to its applicability under transfer-learning conditions, specifically targeting a single type of degradation. The role of the decoders is limited to classifying the input images into two categories: clean and degraded. Given that the paired data provides an equal number of samples for both categories, there is no issue of class imbalance, which permits the use of Cross Entropy Loss instead of Focal Loss.

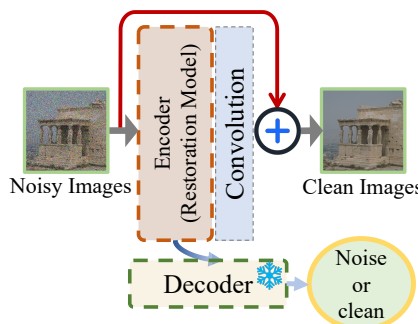

Figure 3: DC-guided training is used for cross-degradation transfer learning. The target task in this figure is denoising.

## 4 EXPERIMENTS

We evaluate our DCPT in four different settings. **(1) All-in-One**: A single model is fine-tuned after DCPT to perform image restoration across multiple degradation. Following previous state-of-the-art works (Zhang et al., 2023; Luo et al., 2023b), we evaluate on five restoration (5D) tasks and 10D tasks. **(2) Single-task**: We report the performance on unseen degradation of all-in-one trained models without fine-tuning following (Zhang et al., 2023; Ai et al., 2024).To highlight DCPT's impact on single-task pre-training, we present fine-tuning results of DCPT pre-trained models in specific single-task settings. **(3) Mixed degradation**: We evaluate the fine-tuned model under mixed degradation to verify whether DCPT is suitable for the restoration of complex mixed-degraded images, such as composite weather. **(4) Transfer learning**: We evaluate the transfer learning capability of restoration models trained by DC-guided training or not between different image restoration tasks. The efficacy of DC-guided training in enhancing the network's cross-task transfer ability is demonstrated.

### 4.1 IMPLENTATION DETAILS.

For the 5/10D all-in-one restoration task, datasets comprising five or ten types of degradation are employed to execute DCPT. In contrast, for other downstream restoration tasks, a uniform approach utilizing datasets with ten types of degradation is adopted for DCPT. We use PSNR and SSIM in the sRGB color space as distortion metrics. We also give the training and dataset details for each task in Appendix B. Due to space constraints, all numerical results for methods under 10D all-in-one and mixed degradation are presented in Appendix C.

### 4.2 ALL-IN-ONE IMAGE RESTORATION

We first assess the performance gain of DCPT on different networks in all-in-one image restoration.

**5D all-in-one image restoration results** with classic image restoration networks (Liang et al., 2021; Zamir et al., 2022; Chen et al., 2022) are reported in Table 2 and Figure 4. It can be observed that regardless of whether the network is CNN or Transformer, whether it is a straight network or a UNet-like network, DCPT consistently achieves an average performance improvement of **2.08 dB and above** on the 5D all-in-one image restoration. This indicates that DCPT is compatible with a wide range of network architectures. In comparison to multi-stage methods, DCPT also demonstrates consistent performance improvement. When Restormer is taken as the basic model, the IDR is only improved by 0.74 dB compared to its base method, while DCPT can give an performance gain of **2.44 dB**. It is also important to note that IDR requires 1200 epochs training, whereas DCPT only necessitates 20 epochs pre-training and 50 epochs fine-tuning.

| Method | Dehazing | Deraining | Denoising | Deblurring | Low-Light | Average |
|---|---|---|---|---|---|---|
| | PSNR↑/SSIM↑ | PSNR↑/SSIM↑ | PSNR↑/SSIM↑ | PSNR↑/SSIM↑ | PSNR↑/SSIM↑ | PSNR↑/SSIM↑ |
| AirNet | 21.04 / 0.884 | 32.98 / 0.951 | 30.91 / 0.882 | 24.35 / 0.781 | 18.18 / 0.735 | 25.49 / 0.846 |
| IDR | 25.24 / 0.943 | 35.63 / 0.965 | 31.60 / 0.887 | 27.87 / 0.846 | 21.34 / 0.826 | 28.34 / 0.893 |
| InstructIR | 27.10 / 0.956 | 36.84 / 0.973 | 31.40 / 0.887 | 29.40 / 0.886 | 23.00 / 0.836 | 29.55 / 0.907 |
| SwinIR | 21.50 / 0.891 | 30.78 / 0.923 | 30.59 / 0.868 | 24.52 / 0.773 | 17.81 / 0.723 | 25.04 / 0.835 |
| **DCPT-SwinIR** | **28.67 / 0.973** | **35.70 / 0.974** | **31.16 / 0.882** | **26.42 / 0.807** | **20.38 / 0.836** | **28.47 / 0.894** |
| NAFNet | 25.23 / 0.939 | 35.56 / 0.967 | 31.02 / 0.883 | 26.53 / 0.808 | 20.49 / 0.809 | 27.76 / 0.881 |
| **DCPT-NAFNet** | **29.47 / 0.971** | **35.68 / 0.973** | **31.31 / 0.886** | **29.22 / 0.883** | **23.52 / 0.855** | **29.84 / 0.914** |
| Restormer | 24.09 / 0.927 | 34.81 / 0.962 | 31.49 / 0.884 | 27.22 / 0.829 | 20.41 / 0.806 | 27.60 / 0.881 |
| **DCPT-Restormer** | **29.86 / 0.973** | **36.68 / 0.975** | **31.46 / 0.888** | **28.95 / 0.879** | **23.26 / 0.842** | **30.04 / 0.911** |
| PromptIR | 25.20 / 0.931 | 35.94 / 0.964 | 31.17 / 0.882 | 27.32 / 0.842 | 20.94 / 0.799 | 28.11 / 0.883 |
| **DCPT-PromptIR** | **30.72 / 0.977** | **37.32 / 0.978** | **31.32 / 0.885** | **28.84 / 0.877** | **23.35 / 0.840** | **30.31 / 0.911** |

Table 2: *5D all-in-one image restoration results.* All the classic architectures pre-trained with DCPT outperform the methods that require two-stage training and external degradation identifying module (AirNet (Li et al., 2022) and IDR (Zhang et al., 2023)).

**10D all-in-one image restoration results.** Furthermore, the degradation types are scaled up to 10 to ascertain the efficacy of DCPT in the presence of a greater number of degradation types. Following the DACLIP (Luo et al., 2023b) and InstructIR (Marcos V. Conde, 2024), we use NAFNet as the basic restoration model because its concision. The averaged performance of the restoration model under 10 degradation is presented in Table 3. It can be observed that, in comparison to abstract CLIP embeddings (Luo et al., 2023b), complex human instruct (Marcos V. Conde, 2024; Guo et al., 2025),

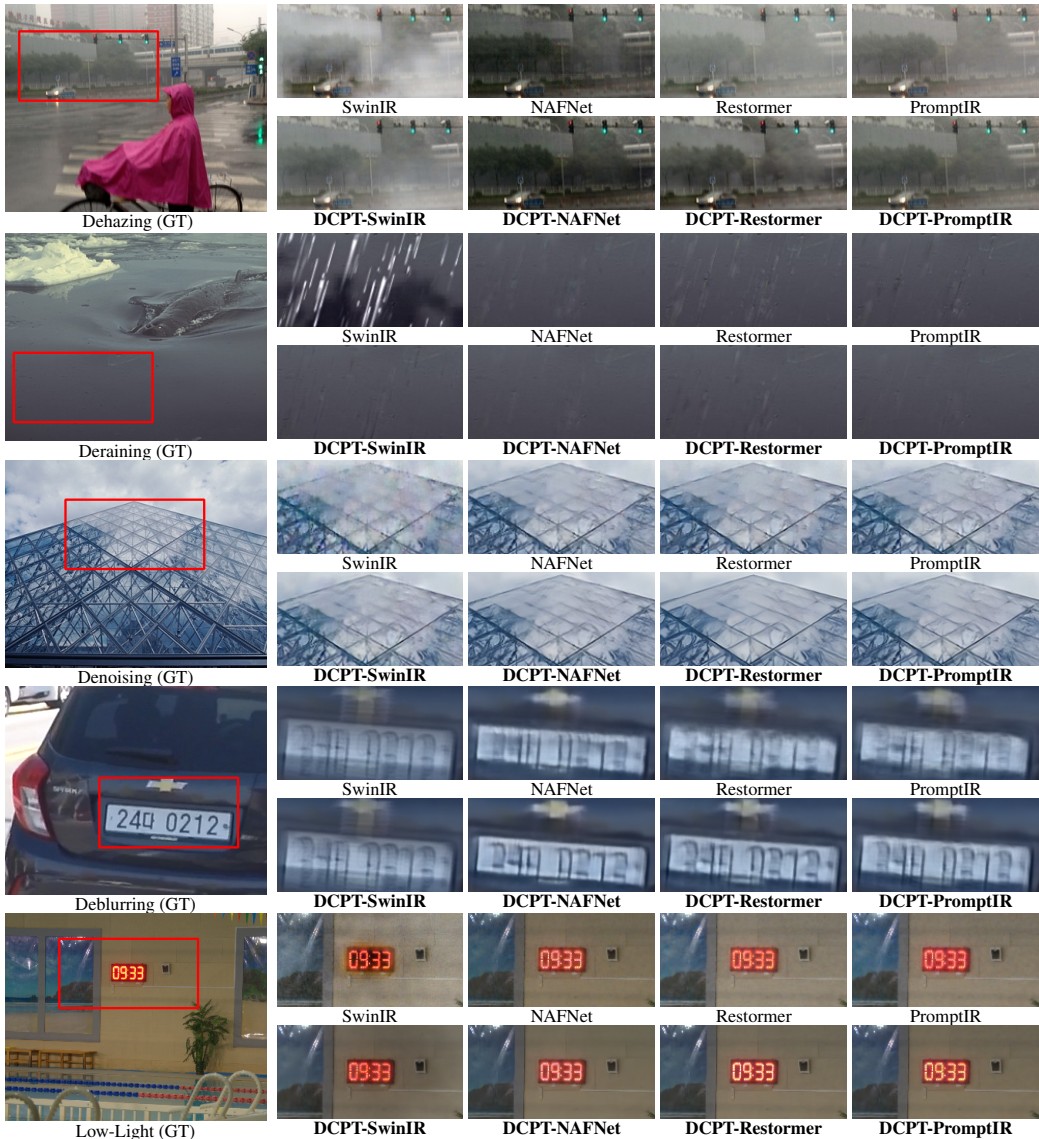

Figure 4: *Visual comparison on 5D all-in-one image restoration datasets.* Zoom in for best view.

and large diffusion model (Zheng et al., 2024), the latent degradation classification prior of the model trained with DCPT is more effective in addressing the all-in-one restoration task. This conclusion can be substantiated by the superior performance gain (**2.55 dB** in PSNR) of NAFNet trained with DCPT on the 10D all-in-one restoration task. We also provide a radar chart to show that DCPT-NAFNet outperforms existing all-in-one restoration models across all tasks. The specific metric values have been provided in Appendix C.2.

### 4.3 SINGLE-TASK IMAGE RESTORATION

A further analysis is conducted to determine the suitability of DCPT for single-task image restoration pre-training from two perspectives. **i. Zero-shot (ZS)**: This assesses whether models trained with DCPT under *5D all-in-one* fine-tuning are used to solve single tasks without optimization. **ii. Fine-tuning (FT)**: This assesses whether the model weights pre-trained with DCPT can be directly used for fine-tuning on single-task image restoration. The degradation classifier guided method is not used in single-task fine-tuning for a fair comparison.

**[ZS] Gaussian Denoising.** Table 4 reports the Gaussian denoising results of image restoration models pre-trained with DCPT at different noise levels. It can be observed that the DCPT pre-trained model has achieved notable improvements in all networks and all datasets. This is particularly evident on the

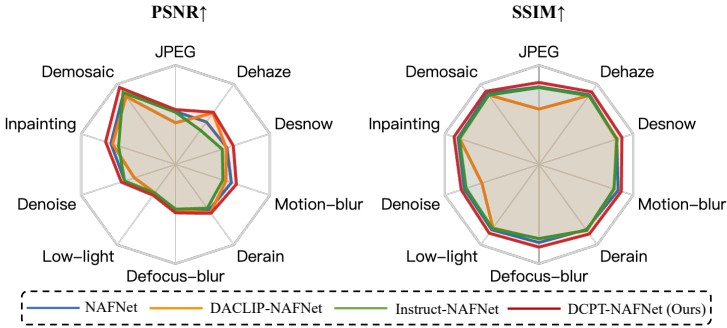

Figure 5: The radar chart of *10D all-in-one image restoration results.*

| Method | 10D-Average PSNR↑/SSIM↑ |
|---|---|
| AirNet | 26.41 / 0.842 |
| TransWeather | 22.83 / 0.779 |
| WeatherDiff | 24.60 / 0.793 |
| PromptIR | 27.93 / 0.851 |
| DiffUIR-L | 28.75 / 0.869 |
| NAFNet | 27.17 / 0.837 |
| +DACLIP | 27.42 / 0.798 |
| +Instruct | 28.30 / 0.862 |
| **+DCPT (Ours)** | **29.72 / 0.888** |

Table 3: *Averaged 10D all-in-one image restoration results.*

high-resolution dataset Urban100 (Huang et al., 2015). In this context, DCPT-SwinIR demonstrates an improvement of **1.11 dB** compared to SwinIR on Gaussian denoising with $\sigma = 50$.

| Method | Urban100 $\sigma=15$ | $\sigma=25$ | $\sigma=50$ | Kodak24 $\sigma=15$ | $\sigma=25$ | $\sigma=50$ | BSD68 $\sigma=15$ | $\sigma=25$ | $\sigma=50$ |
|---|---|---|---|---|---|---|---|---|---|
| AirNet | 33.16 | 30.83 | 27.45 | 34.14 | 31.74 | 28.59 | 33.49 | 30.91 | 27.66 |
| IDR | 33.82 | 31.29 | 28.07 | 34.78 | 32.42 | 29.13 | 34.11 | 31.60 | 28.14 |
| SwinIR | 32.79 | 30.18 | 26.52 | 33.89 | 31.32 | 27.93 | 33.31 | 30.59 | 27.13 |
| **DCPT-SwinIR** | **33.64** | **31.14** | **27.63** | **34.63** | **32.11** | **28.86** | **33.82** | **31.16** | **27.86** |
| NAFNet | 33.14 | 30.64 | 27.20 | 34.27 | 31.80 | 28.62 | 33.67 | 31.02 | 27.73 |
| **DCPT-NAFNet** | **33.64** | **31.23** | **27.98** | **34.72** | **32.28** | **29.21** | **33.94** | **31.31** | **28.12** |
| Restormer | 33.72 | 31.26 | 28.03 | 34.78 | 32.37 | 29.08 | 34.03 | 31.49 | 28.11 |
| **DCPT-Restormer** | **34.14** | **31.79** | **28.58** | **34.96** | **32.49** | **29.40** | **34.09** | **31.46** | **28.25** |
| PromptIR* | 33.27 | 30.85 | 27.41 | 34.44 | 31.95 | 28.71 | 33.85 | 31.17 | 27.89 |
| **DCPT-PromptIR** | **33.88** | **31.49** | **28.15** | **34.78** | **32.30** | **29.14** | **33.96** | **31.32** | **28.08** |

Table 4: **[ZS]** *Gaussian Denoising results* on Urban100, Kodak24 and BSD68 datasets in terms of PSNR ↑.

| Method | UIEB PSNR ↑ | SSIM ↑ |
|---|---|---|
| AirNet | 15.46 | 0.745 |
| IDR | 15.58 | 0.762 |
| SwinIR | 15.31 | 0.740 |
| **DCPT-SwinIR** | **15.78** | **0.774** |
| NAFNet | 15.42 | 0.744 |
| **DCPT-NAFNet** | **15.67** | **0.773** |
| Restormer | 15.46 | 0.745 |
| **DCPT-Restormer** | **15.79** | **0.774** |
| PromptIR* | 15.48 | 0.748 |
| **DCPT-PromptIR** | **15.78** | **0.772** |

Table 5: **[ZS]** *Under-water Enhancement results.*

**[ZS] Under-water Enhancement.** Table 5 presents the results of the underwater image enhancement. In comparison to the base methods, the image restoration models pre-trained with DCPT exhibited an average performance increase of 0.25 ~ 0.47 dB, indicating that DCPT has the capacity for generalization in the unseen degradation.

| Dataset | Method | Deblur-GAN | Deblur-GANv2 | SRN | DMPHN | Restormer | Restormer +DegAE | Restormer **+DCPT (Ours)** |
|---|---|---|---|---|---|---|---|---|
| **GoPro** | PSNR ↑ | 28.70 | 29.55 | 30.26 | 31.20 | 32.92 | 33.03 (**+0.11**) | 33.12 (**+0.20**) |
| | SSIM ↑ | 0.858 | 0.934 | 0.934 | 0.940 | 0.961 | - | 0.962 (**+0.01**) |
| **HIDE** | PSNR ↑ | 24.51 | 26.62 | 28.36 | 29.09 | 31.22 | 31.43 (**+0.21**) | 31.47 (**+0.25**) |
| | SSIM ↑ | 0.871 | 0.875 | 0.915 | 0.924 | 0.942 | - | 0.946 (**+0.04**) |

Table 6: **[FT]** *Single Image Motion Deblurring results* in the single-task setting on GoPro dataset. DCPT outperforms DegAE (Liu et al., 2023a) on the image motion deblurring task pre-training.

**[FT] Single Image Motion Deblurring.** Table 6 shows that Restormer pre-trained with DCPT outperform 0.2 dB on GoPro (Nah et al., 2017). DCPT is suitable for pre-training under single task.

## 4.4 IMAGE RESTORATION ON MIXED DEGRADATION

**Results of restoration on mixed degradation** are displayed in Table 7 to demonstrate that DCPT can also deliver substantial performance enhancements to the restoration model in mixed degradation scenarios. We performed experiments on CDD11 (Guo et al., 2025), a mixed degradation restoration benchmark. NAFNet pre-trained with DCPT achieves the highest performance.

Models pre-trained with DCPT robustly reconstruct HQ images in various mixed degradation by incorporating latent degradation classification prior to image restoration models. When employing NAFNet as the restoration backbone, DCPT demonstrates a performance enhancement of **5.20 dB** for mixed degradations involving low-light, haze, and rain degradation, and **5.58 dB** for those involving low-light, haze, and snow degradation. For compelling evidence, Figure 6 provides the visual comparison of image restoration in three composite degradation samples (low-light + haze + rain). NAFNet pre-trained with DCPT can restore more natural result from mixed-degraded image and fully preserve image texture and detail such as lighting.

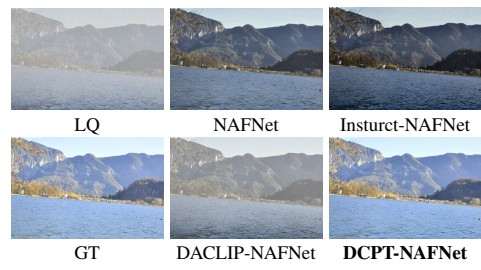

LQ    NAFNet    Insturct-NAFNet

GT    DACLIP-NAFNet    **DCPT-NAFNet**

Figure 6: *Visual comparison on low-light + haze + rain samples.* The DCPT enables the NAFNet to restore HQ images from mixed degradation while adjusting lighting to realistic conditions. In contrast, neither CLIP nor human-instruct can achieve both tasks concurrently. Zoom in for best view.

| Method | low-light+haze+rain PSNR↑/SSIM↑ | low-light+haze+snow PSNR↑/SSIM↑ |
|---|---|---|
| AirNet | 21.80 / 0.708 | 22.23 / 0.725 |
| TransWeather | 21.55 / 0.678 | 21.01 / 0.655 |
| WeatherDiff | 21.23 / 0.716 | 21.03 / 0.698 |
| PromptIR | 23.74 / 0.752 | 23.33 / 0.747 |
| DiffUIR-L | 25.46 / 0.799 | 25.89 / 0.802 |
| OneRestore | 25.18 / 0.795 | 25.28 / 0.797 |
| NAFNet | 21.03 / 0.682 | 20.82 / 0.690 |
| +DACLIP | 22.96 / 0.712 | 23.32 / 0.751 |
| +Instruct | 24.84 / 0.777 | 24.32 / 0.760 |
| **+DCPT (Ours)** | **26.23 / 0.805** | **26.40 / 0.807** |

Table 7: *Mixed degraded image restoration results* on CDD (Guo et al., 2025) dataset.

## 4.5 TRANSFER LEARNING CROSS DIFFERENT DEGRADATION

We are curious about whether the degradation classifiers help model generalization across different degradation. We run these experiments to study the cross-task generalization of image restoration models with or without DC-guided training, e.g., Restormer (Zamir et al., 2022).

| DC-guided | Target task | Denoise | | Deblur | | Derain | |
|---|---|---|---|---|---|---|---|
| | Source task | Deblur | Derain | Denoise | Derain | Denoise | Deblur |
| ✘ | PSNR ↑ | 31.50 | 31.65 | 25.44 | 27.51 | 31.99 | 32.85 |
| ✔ | PSNR ↑ | 31.62 | 31.69 | 30.36 | 28.79 | 36.29 | 35.77 |
| Supervised | PSNR ↑ | 31.78 | | 32.92 | | 36.74 | |

Table 8: *Transfer learning results* with and without degradation classifier (DC) guided. "Supervised" means the model is randomly initialized and trained without the DC-guided training method.

**Transfer learning results** are shown in Table 8. The results of the non-DC-guided experiments presents that *image restoration model is hard to generalize cross different degradation.* In comparison to random initialization, the performance of the model is reduced by 7.48 dB in the image motion deblurring and 4.75 dB in the image deraining when the model is initialized with a trained model on the Gaussian denoising.

However, the DC-guided model shows higher performance in cross-task transfer learning. When model is guided by DC, the reduction is down to 2.56 dB in the image motion deblurring and 0.45 dB in the image deraining when the model is initialized with a trained model on the Gaussian denoising. This suggests that DC has the ability to help image restoration models generalize across tasks.

DC-guided training achieves greater performance gains on tasks that are more difficult to generalize to. To illustrate, the model trained for deraining can achieve a 1.28 dB gain after DC guidance, while the model trained for denoising can achieve a 4.92 dB gain after DC-guided. However, the former has superior generalization performance without DC-guided. This phenomenon is of considerable interest. The cross-task generalization ability of image restoration models may be worth studying.

## 4.6 ABLATION STUDIES

We demonstrate the necessity of decoder architecture, multi-scale feature extraction, and training stages in DCPT through the performance of several ablation experiments. These experiments are performed with PromptIR (Potlapalli et al., 2023) on the 5D all-in-one image restoration task.

**Impact of the decoder architecture.** Table 9 illustrates that the decoder architecture can impact the pre-training performance of DCPT. When Vision Transformer is used as the decoder, DCPT achieves better pre-training effects. Investigating the design of DCPT decoders for various restoration models presents an intriguing direction for future research.

**Impact of multi-level feature extraction.** Table 10 shows that multi-level feature extraction can provide further performance gains for models pre-trained with DCPT.

**Impact of different training stages.** Table 11 demonstrates that combining degradation classification stage ($S_{DC}$) and generation stage ($S_G$) can optimize the model's performance. If $S_{DC}$ is the only

| Decoder | Dehazing | Deraining | Denoising | Debluring | Low-light |
| --- | --- | --- | --- | --- | --- |
| | PSNR ↑ | PSNR ↑ | PSNR ↑ | PSNR ↑ | PSNR ↑ |
| ResNet | 30.72 | 37.32 | 31.32 | 28.84 | 23.35 |
| ViT | 30.88 | 37.74 | 31.37 | 28.93 | 23.22 |
| ViM | 30.64 | 37.30 | 31.32 | 28.80 | 23.18 |

Table 9: Ablations of the decoder architecture. The denoising results are calculated on $\sigma = 25$. ResNet is the baseline. ViT is Vision Transformer while ViM is Vision Mamba.

| Multi-level | Dehazing | Deraining | Denoising | Debluring | Low-light |
| --- | --- | --- | --- | --- | --- |
| | PSNR ↑ | PSNR ↑ | PSNR ↑ | PSNR ↑ | PSNR ↑ |
| None | 25.20 | 35.94 | 31.17 | 27.32 | 20.94 |
| ✗ | 30.23 | 36.88 | 31.27 | 28.14 | 23.00 |
| ✔ | 30.72 | 37.32 | 31.32 | 28.84 | 23.35 |

Table 10: Ablations of multi-level feature extraction. The denoising results are calculated on $\sigma = 25$. "None" means the model is trained without DCPT.

training stage of DCPT, restoration models still benefit from it. This suggests that the performance gain brought by DCPT is primarily due to the degradation classification part rather than the generative ability learning. If $S_G$ is the only training stage of DCPT, residual learning (Zhang et al., 2017) will cause the model to an identity function, reducing model's representation ability and contradicting the aim of pre-training.

**Necessity of pre-training.** DC-Train is subjected to ablation to evaluate the necessity of pre-training. The distinction between DC-Train and DCPT lies in DC-Train's direct utilization of paired degraded-clean image pairs for restoration training during the generation stage, eschewing pre-training and fine-tuning for downstream restoration tasks. The results presented in Table 12 demonstrate that DCPT surpasses DC-Train, thereby underscoring the necessity of pre-training.

| $S_{DC}$ | $S_G$ | Dehazing | Deraining | Denoising | Debluring | Low-light |
| --- | --- | --- | --- | --- | --- | --- |
| ✗ | ✗ | 25.20 | 35.94 | 31.17 | 27.32 | 20.94 |
| ✗ | ✔ | 24.88 | 31.79 | 30.04 | 24.59 | 17.32 |
| ✔ | ✗ | 30.22 | 37.08 | 31.21 | 27.96 | 21.22 |
| ✔ | ✔ | 30.72 | 37.32 | 31.32 | 28.84 | 23.35 |

Table 11: Results of different losses in terms of PSNR ↑. The denoising results are calculated on $\sigma = 25$.

| Method | Dehazing | Deraining | Denoising | Debluring | Low-light |
| --- | --- | --- | --- | --- | --- |
| DC-Train | 30.28 | 37.00 | 31.28 | 28.54 | 22.78 |
| DCPT | 30.72 | 37.32 | 31.32 | 28.84 | 23.35 |

Table 12: Ablations of whether pre-training in terms of PSNR ↑. The "DC-Train" method means that degraded-clean image pairs are used directly in the generation stage of DCPT without pre-training and fine-tuning. The denoising results are calculated on $\sigma = 25$.

### 4.7 DISCUSSION

**Discrimination hidden in restoration.** Previous research (Liu et al., 2023b) has investigated the super-resolution model's ability to distinguish between different types of degradation during the restoration process. The preliminary experiment presented in Sec. 3.1 demonstrated that the randomly initialized model is capable of degradation classifying. Furthermore, the application of supervised all-in-one training enhanced the models' ability to classify degradation while also imparting certain generalization capabilities. These results indicate that there is discrimination hidden in restoration.

The results presented in our paper highlight the effectiveness of discriminative prior in pre-training for image restoration. They reveal that incorporating sufficient discriminative information into the model before training can significantly improve its performance. We hypothesize that integrating superior degradation-aware discriminative information into the restoration model and maximizing its discriminative capacity will further enhance its performance. It is anticipated that this hypothesis will pave the way for the development of lots of novel pre-training methods for universal restoration field.

## 5 CONCLUSION

In this paper, we observed that the randomly initialized model exhibited baseline capability in degradation classification, while the model trained in an all-in-one manner demonstrated a remarkably higher degree of accuracy in this regard. Furthermore, this ability exhibited generalization across different contexts. To fully expand and utilize the degradation classification ability of the restoration model, we developed Degradation Classification Pre-Training (DCPT) and confirmed its effectiveness in universal image restoration and transfer learning tasks. Owing to the incorporation of degradation classification prior via DCPT, restoration models pre-trained with this method demonstrate an all-in-one performance improvement surpassing 2 dB and exhibit a performance augmentation exceeding 5 dB under mixed degradation conditions. Further investigation is warranted into the discriminative behavior of the restoration model to the input image. This may potentially lead to the development of more generalized universal image restoration techniques.

**Ethics Statement.** This paper presents work whose goal is to advance the field of image restoration. There are many potential societal consequences of our work. Given the increasing capabilities of image restoration techniques, we advocate avoiding the misuse of related technologies, such as forging misleading images or restoring and enhancing images for malicious purposes.

**Reproducibility Statement.** We state that DCPT is highly reproducible. Appendix A encompasses the experimental details of the pre-experiment in Sec. 3.1 and try to explain why DCPT works. Comprehensive dataset details are delineated in Appendix B.1. Implementation details on our main experiences are provided in Appendix B.2, and the Pytorch-like source code of DCPT is presented in Appendix D. It is anticipated that these supplementary materials can sufficiently demonstrate the reproducibility of DCPT.

## ACKNOWLEDGEMENT

This work was supported financially in part by the Natural Science Foundation of China (62394311, 82371112, 623B2001); in part by the Science Foundation of Peking University Cancer Hospital (JC202505); in part by Natural Science Foundation of Beijing Municipality (Z210008).

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

## A DETAILS OF EXPERIENCE IN MOTIVATION

### A.1 DATASETS

We randomly select 500 images from the listed datasets for the motivation verification experiment, with 100 images randomly chosen for each degradation. Source datasets: Test100L (Yang et al., 2019) for **deraining**; SOTS (Li et al., 2018) for **dehazing**; SIDD (Abdelhamed et al., 2018) for **denoising**; GoPro (Nah et al., 2017) for **motion deblurring**; and LOL (Wei et al., 2018) for **low-light enhancement**.

Once the input images are determined, they are sent to the restoration model to obtain the features. To ensure that the output features have the same dimension, we choose center-cropped images with the resolution of $128 \times 128$. The output features are flattened for subsequent classification using kNN.

### A.2 IMPLEMENTATION DETAILS

**Dataset split.** Before the experiment, we randomly divide the training set and the test set in a ratio of 2:1. We ensure that the data volume of each degradation in the training set and the test set is evenly distributed.

**3D all-in-one trained models.** For evaluating the degradation classification ability of 3D all-in-one trained models, we train image restoration models under all-in-one settings following AirNet (Li et al., 2022).

**kNN settings.** We classify the degradation type using the last block's feature of restoration model. Uniform weight function is used in prediction. KDTree algorithm is used to compute the nearest neighbors. The iteration numbers for this kNN are set to 2k.

### A.3 MORE RESULTS ON PRELIMINARY EXPERIENCE

Based on the motivation in Section 3.1, we conjecture that the significant improvement of DCPT is due to *DCPT can advance the degradation understanding step before restoration training*. We conducted experiments on NAFNet to test this hypothesis. *If it is correct, the model's performance will improve as the initial degradation classification accuracy increases.*

| DCPT iterations | 0 | 25k | 50k | 75k | 100k |
|---|---|---|---|---|---|
| Initial DC Acc. (%) | 52 | 75 | 88 | 93 | 94 |
| PSNR (dB) | 27.76 | 29.32 | 29.67 | 29.84 | 29.84 |

Table 13: NAFNet's performance improved as the initial degradation classification accuracy increased. The PSNR are averaged among 5 tasks in 5D all-in-one restoration.

As shown, NAFNet's performance improved as the initial degradation classification accuracy increased. This supports our conjecture that **one of the core reasons why DCPT works is that it advances the degradation understanding step before restoration training**.

### A.4 ABLATION STUDY ON CLASSIFICATION DATASETS

To demonstrate the capability of restoration models in classifying degradation rather than datasets, we design an ablation study on classification datasets. We used 485 normal-light images from LOL-v1 (Wang et al., 2021) as clean images and applied five degradations: downsample, blur, noise, JPEG, and low-light. The new results are shown below.

| Methods | NAFNet | SwinIR | Restormer | PromptIR |
|---|---|---|---|---|
| Acc. on Random initialized (%) | $33 \pm 5$ | $38 \pm 6$ | $40 \pm 6$ | $34 \pm 4$ |
| Acc. on 3D all-in-one trained (%) | $79 \pm 3$ | $83 \pm 4$ | $85 \pm 3$ | $80 \pm 2$ |

Table 14: Degradation classification accuracy in ablation study on classification datasets. The results are averaged under five random seeds.

It can be seen that Randomly initialized models achieve 33 ~ 40% accuracy in degradation classification. After 3D all-in-one training, accuracy improves to 79% or higher, even for unseen degradations, confirming the assertions in Sec. 3.1.

Go back to Section 3.1.

## B  DETAILS OF MAIN EXPERIENCES

### B.1  DATASET DETAILS

**Dataset for all-in-one setting.** To ensure that the model is not exposed to wider images during the pre-training, resulting in performance gains, the dataset in the all-in-one setting is identical to that in the DCPT, with only slight modifications in the sampler.

For 3D and 5D all-in-one restoration, following AirNet (Li et al., 2022), a combination of various image restoration datasets is employed: Rain200L (Yang et al., 2017), which contains 200 training images for deraining; RESIDE (Li et al., 2018), which contains 72,135 training images and 500 test images (SOTS) for dehazing; BSD400 (Martin et al., 2001b) and WED (Ma et al., 2016), which contain 5,144 training images for Gaussian denoising; GoPro (Nah et al., 2017), which contains 2,103 training images and 1,111 test images for single image motion deblurring; and LOL (Wei et al., 2018), which contains 485 training images and 15 test images for low-light enhancement.

For 10D all-in-one restoration, a combination of various image restoration datasets is employed: RainH&RainL (Yang et al., 2017), which contains 3000 training images for deraining; RESIDE (Li et al., 2018), which contains 72,135 training images and 500 test images (SOTS) for dehazing; Snow100K (Liu et al., 2018) for image desnowing; BSD400 (Martin et al., 2001b) and WED (Ma et al., 2016), which contain 5,144 training images for Gaussian denoising, JPEG, demosaic, and inpainting; GoPro (Nah et al., 2017), which contains 2,103 training images and 1,111 test images for single image motion deblurring; DPDD (Abuolaim & Brown, 2020) for single image defocus deblurring; and LOL (Wei et al., 2018), which contains 485 training images and 15 test images for low-light enhancement.

We use repeated sampler for all-in-one dataset following (Li et al., 2022; Zhang et al., 2023).

**Dataset Sampler in all-in-one setting.** For degradation with lesser training data, such as image deraining, we use repeat sampler technology to ensure that there are enough training pairs for each degradation. The repeat ratio is [1H, 120R, 9N], where H, R, and N represent dehaze, derain, and denoise, respectively. For the 5D all-in-one image restoration, the datasets for dehazing, deraining, Gaussian denoising, motion deblurring, and low-light enhancement are integrated for fine-tuning. The repetition ratio is [1H, 300R, 15N, 5B, 60L], where the H, R, N, B, and L represent dehaze, derain, denoise, deblur, and low-light enhancement, respectively.

**Dataset for single-task setting.** In single-task setting, for zero-shot (**ZS**) settings, we evaluate the 5D all-in-one models on Urban100 Huang et al. (2015), Kodak24 (Franzen, 2013) and BSD68 (Martin et al., 2001a) for Gaussian denoising; and UIEB (Li et al., 2019) for under-water image enhancement, which is a training-unseen task. For fine-tune (**FT**) settings, we train Restormer (Zamir et al., 2022) on GoPro (Nah et al., 2017) for single image motion deblurring for a fair comparison with DegAE (Liu et al., 2023a).

**Dataset for mixed degradation setting.** We use CDD training and test datasets for testing on mixed degradation setting. CDD encompassing 11 categories of image degradations and their clear counterparts. These degraded samples include low (low-light), haze, rain, snow, low+haze, low+rain, low+snow, haze+rain, haze+snow, low+haze+rain, and low+haze+snow. There are 1,383 high-resolution clear images for producing 11 composite degradations. The overall dataset is split into 13,013 image pairs for training and 2,200 for testing.

**Dataset for transfer-learning setting.** Following Restormer (Zamir et al., 2022), the training datasets employed are DF2K (Agustsson & Timofte, 2017), WED (Ma et al., 2016) and BSD400 (Martin et al., 2001b) for Gaussian denoising; GoPro (Nah et al., 2017) for motion deblurring; and Rain13K (Chen et al., 2021b) for image deraining. The denoising, deblurring and deraining results are tested on the BSD68 (Martin et al., 2001a), GoPro (Nah et al., 2017) testset and Rain100L (Yang et al., 2019), respectively.

## B.2 IMPLEMENTATION DETAILS

**Implementation details for DCPT.** During DCPT, image restoration models (encoder) and degradation classifiers (decoder) are all trained by AdamW (Kingma & Ba, 2014) with no weight decay for 100k iters with batch-size 32 on $128 \times 128$ image patches on 4 NVIDIA L40 GPUs. Due to the heterogeneous encoder-decoder design, we employ distinct learning rates for encoder and decoder. The learning rate is set to $3 \times 10^{-4}$ for encoder and $1 \times 10^{-4}$ for decoder. The learning rate does not alter during the DCPT. After DCPT, the parameters of encoder will be used to initialize the image restoration models.

**Implementation details in all-in-one setting.** We initialize the image restoration models using the parameters pre-trained by DCPT. For fairness and convenience, we adopt the same training policy for different backbones. We use the AdamW (Kingma & Ba, 2014) optimizer with the initial learning rate $3 \times 10^{-4}$ gradually reduced to $1 \times 10^{-6}$ with the cosine annealing schedule to train our image restoration models. The training runs for 750k iters with batch size 32 on 4 NVIDIA L40 GPUs.

**Implementation details for fine-tuning in single-task setting.** The training hyper-parameters employed are identical to those utilized by Restormer (Zamir et al., 2022). The sole distinction is that we use the DCPT pre-trained parameters to initialize the model. The fine-tuning runs on 1 NVIDIA A100 GPU.

**Implementation details for fine-tuning in mixed degradation setting.** We initialize the NAFNet using the parameters pre-trained by DCPT on 10D all-in-one restoration datasets. We use the AdamW (Kingma & Ba, 2014) optimizer with the initial learning rate $3 \times 10^{-4}$ gradually reduced to $1 \times 10^{-6}$ with the cosine annealing schedule to train our image restoration models on CDD training dataset. The training runs for 750k iters with batch size 32 on 4 NVIDIA L40 GPUs.

**Implementation details for transfer learning.** The initialization of the image restoration models is contingent upon the parameters that have been trained from the source task. The image restoration models are trained on the source task dataset with 100k iterations using a learning rate of $3 \times 10^{-4}$ and batch size of 8. When DC guides the restoration model to execute cross-task transfer learning, we add $L_{cls}$ into the loss function to help the model discern the degradation of the input images. We use the DC generated by DCPT in 5D all-in-one image restoration task. DC's parameters are frozen during the transfer learning. The fine-tuning runs on 1 NVIDIA A100 GPU.

## C MORE RESULTS ON MAIN EXPERIENCES

### C.1 3D ALL-IN-ONE IMAGE RESTORATION

**3D all-in-one image restoration results** are reported in Table 15. PromptIR (Potlapalli et al., 2023), an efficient image restoration network designed for all-in-one tasks, achieves an average performance gain of **0.81 dB** after DCPT. It is worth mentioning that PromptIR after DCPT improves performance across all tasks. For example, DCPT can provide a **2.06 dB** gain on the image deraining task, and a **1.33 dB** gain on the outdoor image dehazing task.

| Method | Dehazing on SOTS | Deraining on Test100L | Denoising on BSD68 | | | Average |
| | | | $\sigma = 15$ | $\sigma = 25$ | $\sigma = 50$ | |
| | PSNR↑/SSIM↑ | PSNR↑/SSIM↑ | PSNR↑/SSIM↑ | PSNR↑/SSIM↑ | PSNR↑/SSIM↑ | PSNR↑/SSIM↑ |
|---|---|---|---|---|---|---|
| BRDNet | 23.23 / 0.895 | 27.42 / 0.895 | 32.26 / 0.898 | 29.76 / 0.836 | 26.34 / 0.693 | 27.80 / 0.843 |
| LPNet | 20.84 / 0.828 | 24.88 / 0.784 | 26.47 / 0.778 | 24.77 / 0.748 | 21.26 / 0.552 | 23.64 / 0.738 |
| FDGAN | 24.71 / 0.929 | 29.89 / 0.933 | 30.25 / 0.910 | 28.81 / 0.868 | 26.43 / 0.776 | 28.02 / 0.883 |
| MPRNet | 25.28 / 0.955 | 33.57 / 0.954 | 33.54 / 0.927 | 30.89 / 0.880 | 27.56 / 0.779 | 30.17 / 0.899 |
| DL | 26.92 / 0.931 | 32.62 / 0.931 | 33.05 / 0.914 | 30.41 / 0.861 | 26.90 / 0.740 | 29.98 / 0.876 |
| AirNet | 27.94 / 0.962 | 34.90 / 0.968 | 33.92 / 0.933 | 31.26 / 0.888 | 28.00 / 0.797 | 31.20 / 0.910 |
| PromptIR | 30.58 / 0.974 | 36.37 / 0.972 | 33.98 / 0.933 | 31.31 / 0.888 | 28.06 / 0.799 | 32.06 / 0.913 |
| **DCPT-PromptIR** | **31.91 / 0.981** | **38.43 / 0.983** | **34.17 / 0.933** | **31.53 / 0.889** | **28.30 / 0.802** | **32.87 / 0.918** |

Table 15: *3D all-in-one image restoration results.* DCPT outperforms previous all-in-one methods on all tasks, achieving an average performance gain of **0.81dB** compared to its base method PromptIR (Potlapalli et al., 2023).

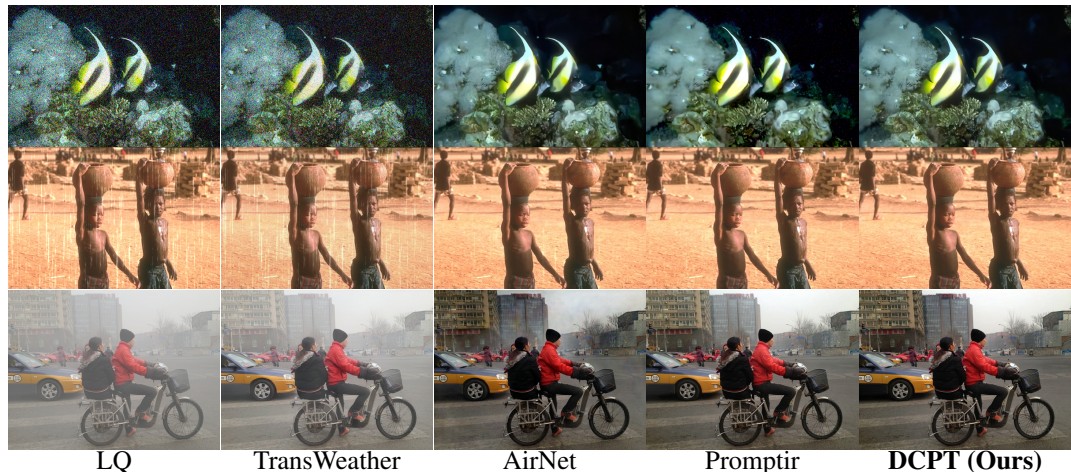

| LQ | TransWeather | AirNet | Promptir | **DCPT (Ours)** |

Figure 7: ***Visual comparison on 3D all-in-one image restoration datasets.*** Top row: Gaussian color denoising on BSD68 (Martin et al., 2001a). Middle row: Image deraining on Test100L (Yang et al., 2019). Bottom row: Image dehazing on SOTS (Li et al., 2018). DCPT-PromptIR can remove the degradation while avoiding the incorrect reconstruction of detailed textures.

## C.2    10D ALL-IN-ONE IMAGE RESTORATION

Here we provide a more detailed comparison between our method with other approaches for 10D all-in-one image restoration. The results on PSNR and SSIM are shown in Table 16 and Table 17.

| PSNR (dB) ↑ | Denoise | Motion-blur | Defocus-blur | JPEG | Dehaze | Desnow | Derain | Demosaic | Low-light | Inpainting | Average |
|---|---|---|---|---|---|---|---|---|---|---|---|
| AirNet | 27.51 | 26.25 | 24.87 | 26.98 | 23.56 | 24.87 | 28.45 | 37.22 | 14.24 | 30.15 | 26.41 |
| NAFNet | 27.16 | 26.12 | 24.66 | 26.81 | 24.05 | 25.94 | 27.32 | 38.45 | 22.16 | 29.03 | 27.17 |
| PromptIR | 27.56 | 26.50 | 25.10 | 26.95 | 25.19 | 27.23 | 29.04 | 38.32 | 23.14 | 30.22 | 27.93 |
| DACLIP-NAFNet | 24.28 | 27.03 | 24.98 | 23.86 | 28.19 | 27.12 | 28.94 | 36.72 | 22.09 | 30.94 | 27.42 |
| InstructIR | 27.13 | 28.70 | 25.33 | 27.02 | 26.90 | 27.35 | 28.11 | 38.18 | 22.81 | 31.48 | 28.30 |
| **DCPT-NAFNet (Ours)** | **28.16** | **30.29** | **25.68** | **27.56** | **30.45** | **29.30** | **29.17** | **40.53** | **22.87** | **33.24** | **29.72** |

Table 16: Comparison of our method with other all-in-one image restoration approaches in terms of PSNR on 10D all-in-one restoration task.

| SSIM ↑ | Denoise | Motion-blur | Defocus-blur | JPEG | Dehaze | Desnow | Derain | Demosaic | Low-light | Inpainting | Average |
|---|---|---|---|---|---|---|---|---|---|---|---|
| AirNet | 0.769 | 0.805 | 0.772 | 0.783 | 0.916 | 0.846 | 0.867 | 0.972 | 0.781 | 0.911 | 0.842 |
| NAFNet | 0.768 | 0.804 | 0.726 | 0.780 | 0.926 | 0.869 | 0.848 | 0.943 | 0.809 | 0.901 | 0.837 |
| PromptIR | 0.774 | 0.815 | 0.704 | 0.784 | 0.933 | 0.887 | 0.876 | 0.992 | 0.829 | 0.918 | 0.851 |
| DACLIP-NAFNet | 0.569 | 0.810 | 0.731 | 0.540 | 0.965 | 0.859 | 0.854 | 0.946 | 0.809 | 0.894 | 0.798 |
| InstructIR | 0.799 | 0.842 | 0.789 | 0.791 | 0.952 | 0.860 | 0.833 | 0.988 | 0.836 | 0.933 | 0.862 |
| **DCPT-NAFNet (Ours)** | **0.799** | **0.900** | **0.810** | **0.800** | **0.972** | **0.912** | **0.878** | **0.990** | **0.858** | **0.965** | **0.888** |

Table 17: Comparison of our method with other all-in-one image restoration approaches in terms of SSIM on 10D all-in-one restoration task.

## C.3    MIXED DEGRADATION DATASET

Here we provide a more detailed comparison between our method with other approaches for mixed degradation image restoration on CDD test dataset. The results on PSNR and SSIM are shown in Table 18 and Table 19.

| PSNR (dB) ↑ | l | h | r | s | l+h | l+r | l+s | h+r | h+s | l+h+r | l+h+s |
|---|---|---|---|---|---|---|---|---|---|---|---|
| NAFNet | 24.50 | 25.34 | 29.24 | 29.54 | 21.91 | 22.75 | 22.79 | 23.67 | 23.86 | 21.03 | 20.82 |
| AirNet | 24.83 | 24.21 | 26.55 | 26.79 | 23.23 | 22.82 | 23.29 | 22.21 | 23.29 | 21.80 | 22.24 |
| TransWeather | 23.39 | 23.95 | 26.69 | 25.74 | 22.24 | 22.62 | 21.80 | 23.10 | 22.34 | 21.55 | 21.01 |
| WeatherDiff | 23.58 | 21.99 | 24.85 | 24.80 | 21.83 | 22.69 | 22.12 | 21.25 | 21.99 | 21.23 | 21.04 |
| PromptIR | 26.32 | 26.10 | 31.56 | 31.53 | 24.49 | 25.05 | 24.51 | 24.54 | 23.70 | 23.74 | 23.33 |
| OneRestore | 26.55 | 32.71 | 33.48 | 34.50 | 26.15 | 25.83 | 25.56 | 30.27 | 30.46 | 25.18 | 25.28 |
| InstructIR | 26.70 | 32.61 | 33.51 | 34.45 | 24.36 | 25.41 | 25.63 | 28.80 | 29.64 | 24.84 | 24.32 |
| DACLIP-NAFNet | 26.86 | 33.09 | 33.91 | 35.29 | 25.82 | 25.74 | 26.08 | 29.36 | 29.75 | 22.96 | 23.32 |
| DCPT-NAFNet (Ours) | **27.61** | **36.71** | **35.75** | **37.92** | **27.15** | **26.75** | **26.70** | **32.63** | **33.40** | **26.23** | **26.40** |

Table 18: Comparison of our method with other universal image restoration approaches in terms of PSNR on mixed degradation restoration task (CDD dataset). In this table's header, "l" stands for low-light, "h" stands for haze, "r" stands for rain, and "s" stands for snow.

| SSIM ↑ | l | h | r | s | l+h | l+r | l+s | h+r | h+s | l+h+r | l+h+s |
|---|---|---|---|---|---|---|---|---|---|---|---|
| NAFNet | 0.736 | 0.960 | 0.899 | 0.931 | 0.747 | 0.665 | 0.676 | 0.871 | 0.904 | 0.682 | 0.690 |
| AirNet | 0.778 | 0.951 | 0.891 | 0.919 | 0.779 | 0.710 | 0.723 | 0.868 | 0.901 | 0.708 | 0.725 |
| TransWeather | 0.725 | 0.924 | 0.899 | 0.890 | 0.721 | 0.694 | 0.661 | 0.876 | 0.868 | 0.678 | 0.655 |
| WeatherDiff | 0.763 | 0.904 | 0.885 | 0.888 | 0.756 | 0.730 | 0.707 | 0.868 | 0.868 | 0.716 | 0.698 |
| PromptIR | 0.805 | 0.969 | 0.946 | 0.960 | 0.789 | 0.771 | 0.761 | 0.924 | 0.925 | 0.752 | 0.747 |
| OneRestore | 0.827 | 0.991 | 0.964 | 0.974 | 0.829 | 0.803 | 0.797 | 0.960 | 0.966 | 0.795 | 0.797 |
| InstructIR | 0.809 | 0.978 | 0.940 | 0.948 | 0.800 | 0.782 | 0.778 | 0.921 | 0.959 | 0.777 | 0.760 |
| DACLIP-NAFNet | 0.803 | 0.984 | 0.957 | 0.958 | 0.811 | 0.793 | 0.780 | 0.949 | 0.960 | 0.712 | 0.751 |
| **DCPT-NAFNet (Ours)** | **0.833** | **0.995** | **0.975** | **0.985** | **0.830** | **0.812** | **0.808** | **0.973** | **0.980** | **0.805** | **0.807** |

Table 19: Comparison of our method with other universal image restoration approaches in terms of SSIM on mixed degradation restoration task (CDD dataset). In this table's header, "l" stands for low-light, "h" stands for haze, "r" stands for rain, and "s" stands for snow.

# D PyTorch-like code of DCPT

To illustrate the simplicity and efficacy of DCPT, we present the PyTorch-like code of DCPT here. We hope that this code will further improve the reproducibility of DCPT.

```python
### train to generate the clean image
encoder.train()
decoder.eval()
optimizer_encoder.zero_grad()
pix_output = encoder(gt, hook=False)

l_total = 0
# pixel loss
if cri_pixel:
    l_pix = cri_pixel(pix_output, gt)
    l_total += l_pix

### train to classify the degradation
decoder.train()
optimizer_decoder.zero_grad()

hook_outputs = encoder(lq, hook=True)
cls_output = decoder(lq, hook_outputs[::-1])

# classification loss
if cri_cls:
    l_cls = cri_cls(cls_output, dataset_idx)
    l_total += l_cls

l_total.backward()
optimizer_encoder.step()
optimizer_decoder.step()
```

# E Future works

We have validated the efficacy of DCPT under a majority of degradation conditions. In the future, we intend to undertake more comprehensive evaluations of DCPT's performance. Specifically, our future investigations will focus on: 1) Assessing DCPT's capability to enhance model performance in more intricate, real-world scenarios, such as those encountered in the wild; and 2) Addressing mixed degradation scenarios directly during the degradation classification phase. Numerous real-world super-resolution studies can inform the former, while the latter may involve framing degradation classification as a multi-target classification problem.

