# OpenReview forum: "Universal Image Restoration Pre-training via Degradation Classification"
_ICLR.cc/2025/Conference — ICLR 2025 Poster_

### Official Review · Reviewer_X95g · 2024-11-01

**Soundness:** 2
**Presentation:** 3
**Contribution:** 2
**Rating:** 5
**Confidence:** 4

**Summary:**

The paper introduces the Degradation Classification Pre-Training (DCPT) framework for universal image restoration. DCPT leverages degradation classification as weak supervision, allowing restoration models to better generalize across various degradation types. By pre-training an encoder to classify degradation types before restoration, DCPT significantly enhances model performance for both all-in-one and mixed degradation scenarios, achieving up to 6.53 dB improvement. The approach is novel in its use of degradation as a pre-training signal, demonstrating strong potential for improving transfer learning in image restoration tasks.

**Strengths:**

1. The proposed Degradation Classification Pre-Training (DCPT) presents a concept by utilizing degradation classification as weak supervision in the pre-training phase. Unlike traditional self-supervised learning approaches, this method fully utilizes the inherent degradation signals present in restoration tasks, enhancing generalizability across various degradation types.
2. The experimental results are well-supported with rigorous comparisons against state-of-the-art methods, demonstrating consistent improvements across different architectures and settings.
3. The paper clearly explains the design of DCPT, from motivation through methodology, complemented by informative diagrams. The detailed ablation studies contribute to understanding the contributions of each module, making the technical aspects easy to follow.

**Weaknesses:**

1. The evaluation mainly focuses on common degradation types. Including more diverse and challenging degradation scenarios could further validate the robustness of DCPT.
2. The approach relies on degradation classification as weak supervision, which may not be readily available or straightforward for certain types of image restoration tasks, potentially limiting its applicability in more complex real-world settings.

**Questions:**

1. The paper proposes a degradation classification pre-training (DCPT) strategy to generalize across degradation types. However, it remains unclear how this model would handle unseen degradations or complex, real-world degradation combinations (e.g., low light combined with blur and noise). Could the authors clarify how their model would cope with such cases?
2. I think the method proposed in the article is extremely similar to AirNet. Can you provide the difference between it and AirNet?

---

### Official Review · Reviewer_3dGp · 2024-11-02

**Soundness:** 2
**Presentation:** 2
**Contribution:** 3
**Rating:** 6
**Confidence:** 5

**Summary:**

This paper starts from the idea that a network trained on an all-in-one task can perform degradation classification and proposes a novel restoration network framework based on a pre-training strategy. Extensive experiments demonstrate that the proposed framework achieves state-of-the-art performance and can be transferred to various networks.

**Strengths:**

1. The experimental results are impressive in both all-in-one and single-task settings.
2. The proposed framework looks novel and interesting. However, some parts need to be explained.

**Weaknesses:**

1. The main issue with the paper is the lack of clarity in the motivation section. In this section, the authors introduce the idea that a model trained in an all-in-one setting can classify degradation. This core idea is relatively simple and widely known. However, the key question is, how does this relate to the concept of a pre-trained model? The core approach of the paper revolves around pre-training a model that can then be fine-tuned to adapt to various tasks. I believe the alignment between the motivation and the core design and concept of the paper needs to be further improved, or additional experiments should be conducted to verify this connection.

2. The network design section is also confusing, lacking the motivation behind each part or proposed module, which needs further analysis. Merely using experimental data to claim "we achieved performance improvement" is insufficient and does not make for an interesting paper. Why is the pre-training divided into two alternating stages? Why not train them simultaneously? Why does the second stage use paired clean images? And is the second stage necessary? What are the reasons behind these choices? The current explanation in the paper is too brief and lacks persuasiveness.

3. The writing of the paper needs further improvement, as there are many typos, particularly with inconsistencies in tense (multiple instances of past and present tense coexisting).

**Questions:**

In my view, the core issue with the paper lies in the writing, especially in clearly articulating the motivations for each part (the motivation for the core framework and the proposed modules). I am satisfied with the experimental section. My current score is 5, but if the authors can significantly improve their writing, I would be willing to raise my score.

---

### Official Review · Reviewer_dFsS · 2024-11-03

**Soundness:** 3
**Presentation:** 2
**Contribution:** 2
**Rating:** 6
**Confidence:** 5

**Summary:**

This paper proposes a new pre-training method, DCPT (Degradation Classification Pre - Training), which enables the model to learn how to classify the degradation types of input images during universal image restoration pre-training. It utilizes the degradation types of input images as weak supervision information, providing new ideas and methods for the field of universal image restoration. The experimental design is comprehensive, and the performance of DCPT is evaluated in different scenarios such as All - In - One, Single - task, Mixed degradation, and Transfer learning, demonstrating the effectiveness and generalization ability of this method to some extent.

**Strengths:**

1.The degradation types are very important for the universal image restoration task. The article proposes the discriminative information regarding degradation, inherent in the image restoration model and verifies this view on multiple models.

2.The article uses the degradation types of input images as a kind of weak supervision information and designs a network with an encoder-decoder structure. After DCPT, both convolutional neural networks (CNNs) and transformers show performance improvements and performance enhancements on multiple tasks.

**Weaknesses:**

1. I agree that the degradation types are very important for the universal image restoration task. The article puts forward the view that there is classification information of degradation types in the image restoration model and conducts preliminary experiments for verification. However, it lacks an in-depth discussion of the root causes and does not analyze in depth whether this phenomenon is generalizable.

2. The manuscript needs to be improved in terms of graphical presentation. For example, (a) and (b) in Figure 3 do not have a clear description, and the connection between them is not presented directly and the reference formatting appears inconsistent.I have a few minor suggestions: (1) Please unify the format of 't-SNE,' as both 'T-SNE' and 't-SNE' appear inconsistently. (2) Consider adjusting the text size in the figures to more closely match the font size of the main text.

3. Although the paper demonstrates the good performance of DCPT in the experimental part, it lacks sufficient discussion on the possible limitations of this method. For example, on certain specific types of image degradation or datasets, whether there may be performance bottlenecks for DCPT and how to further improve it?

4.In the degradation classification pre-training phase,The article does  not make it clear how many categories does the classification decoder predict， For instance, are 3-, 5-, and 10-D trained separately with DCPT, with the classification decoder predicting 3, 5, and 10 categories, respectively.

**Questions:**

1.	Is the degradation category finally output by the encoder in Figure 3 single? How to deal with input images with multiple degradation types? This figure does not demonstrate how the classification results lead to image restoration in the generation stage.

2.	The ablation experiments in Section 4.5 are completed with PromptIR on 3D all - in - one, which is inconsistent with the 5D all - in - one image and 10D all - in - one image in Section 4.1.Why is it designed like this?

3.	Could you clarify what the 'random initialization model' in Table 1 of Section 3.1 refers to?

4.	At line 249, does the degradation classifier-guided training mean fine-tuning? At this stage, the classifier only needs to differentiate between clean and degraded categories, whereas the classification decoder in Figure 4 predicts specific degradation types (e.g., noise). Is there a discrepancy here?

---

### Official Review · Reviewer_pyC6 · 2024-11-04

**Soundness:** 3
**Presentation:** 3
**Contribution:** 3
**Rating:** 8
**Confidence:** 5

**Summary:**

This paper studies the intrinsic classification capabilities of pre-trained universal image restoration models. Based on findings that pre-trained IR models are potentially good degradation classifiers, the authors develop their DCPT to facilitate the pre-training of universal image restoration models. Extensive experiments on four distinct settings have evaluated the effectiveness of DCPT.

**Strengths:**

1.	The motivation of DCPT is clearly stated and analyzed in Section 3.1, with sufficient experiments to hold the authors’ hypothesis.
2.	The experiments in this paper are extensive and comprehensive. The authors not only study the performance gain on the DCPT-facilitated restoration model, but also analyze the potentials of DCPT against transfer learning settings. Additionally, most experiments show promising results with over 1dB PSNR gain.
3.	The writing of the paper is fluent and clear, making it easy to understand.

**Weaknesses:**

1.	In line 141, the authors directly state “classify five degradation types” without necessary explanation. We can only infer the types from Figure 2. It would be better to state the types at the beginning.

**Questions:**

While this is a good paper, I still want to ask a few questions.

1.	I’m wondering about the performance of a “direct” degradation classifier against DCPT. As evidenced in Table 1, a random initialized restoration model has a relatively high classification accuracy. Is this because of the model structure of restoration models? Or, does a random initialized classification model, e.g., Resnet-18/50, have also got a similar phenomenon? How about training a classification model to “directly” classify degradation?
2.	While DCPT achieves better performance under mix-degradation settings, I’m wondering about the classification outputs of DCPT. Is DCPT correctly classifying all types of degradations under the mix-degradation setting? Moreover, how does DCPT perform when facing similar degradations, e.g., gaussian blur, defocus blur, and motion blur?

---

### Public Comment · ~Xiaole_Tang1 · 2024-12-03
**Related Work: Residual-conditioned optimal transport (ICML 2024) and its extension**

Dear Authors,

Thanks for the interesting work DCPT. It is important for the research community to focus on learning how to classify the degradation type of input images so we really appreciate your contributions. We recently published a highly related work at ICML 2024 "Residual-conditioned optimal transport" (RCOT, see https://openreview.net/forum?id=irBHPlknxP) for image restoration and its extension "Degradation-Aware Residual-Conditioned Optimal Transport for Unified Image Restoration" (DA-RCOT, https://arxiv.org/abs/2411.01656), which leverage the inherent degradation semantics in the residuals for implicit task perception. We believe the discussion complements your work and claims about degradation classifiers.

All the best!

---

> ### Author Response · Authors · 2024-12-03
> **Thank you for your appreciation**
>
> Dear Xiaole,
>
> Thank you for your appreciation of our work!
>
> We quickly reviewed the two manuscripts you highlighted. We believe they are related to DCPT, however, DCPT's contribution differs from these two.
>
> - RCOT employs residuals as distinct degradation-specific cues, whereas DA-RCOT incorporates multi-scale residuals into the restoration model to enhance the model's all-in-one restoration capacity.
> - DCPT directly encourages the model to become a degradation classifier, allowing the model to perform superior universal restoration without requiring the infusion of external degradation-specific information.
>
> We are grateful to you for drawing our attention to these two excellent works. We will include them in our related works section.
>
> Thank you again for your appreciation.
>
> Best regards,
>
> Authors.

---

### Meta-Review · Area_Chair_42sa · 2024-12-18

**Metareview:**

This paper propose an effective Degradation Classification Pre-Training for universal image restoration. The provided results demonstrate the effectiveness of the proposed method.

The paper received reviews from four experts. The major concerns include the unclear motivation and limited applications on real-world scenarios.

The area chair finds that the provided rebuttal solves the concerns of reviewers well. Based on the recommendations of reviewers, the paper is accepted.

**Additional Comments On Reviewer Discussion:**

In the discussion stage, the authors solve the concerns of  Reviewers pyC6, dFsS, and 3dGp well. No new concern is raised.

---

### Decision · Program_Chairs · 2025-01-22

Accept (Poster)